# Using Model Calibration to Evaluate Link Prediction in Knowledge Graphs

## Abstract

Link prediction models assign scores to predict new, plausible edges to complete knowledge graphs. In link prediction evaluation, the score of an existing edge (positive) is ranked w.r.t. the scores of its synthetically corrupted counterparts (negatives). An accurate model ranks positives higher than negatives, assuming ascending order. Since the number of negatives are typically large for a single positive, link prediction evaluation is computationally expensive. As far as we know, only one approach has proposed to replace rank aggregations by a distance between sample positives and negatives. Unfortunately, the distance does not consider individual ranks, so edges in isolation cannot be assessed. In this paper, we propose an alternative protocol based on posterior probabilities of positives rather than ranks. A calibration function assigns posterior probabilities to edges that measure their plausibility. We propose to assess our alternative protocol in various ways, including whether expected semantics are captured when using different strategies to synthetically generate negatives. Our experiments show that posterior probabilities and ranks are highly correlated. Also, the time reduction of our alternative protocol is quite significant: more than 77% compared to rank-based evaluation. We conclude that link prediction evaluation based on posterior probabilities is viable and significantly reduces computational costs.

## CCS Concepts

• **General and reference** → *Reliability*; *Evaluation*; • **Computing methodologies** → **Semantic networks**.

## Keywords

Knowledge Graph Embedding, Link Prediction, Model Calibration

**ACM Reference Format:**
Anonymous Author(s). 2024. Using Model Calibration to Evaluate Link Prediction in Knowledge Graphs. In *Proceedings of TheWebConf'24: The ACM Web Conference (TheWebConf'24)*. ACM, New York, NY, USA, 10 pages. https://doi.org/10.1145/nnnnnnn.nnnnnnn

## 1 Introduction

Knowledge graphs contain entities of interest (vertices) and relationships between them (directed, labeled edges) [19]. These graphs enable a focus on concepts rather than strings, so they are at the core of search engines, social networks, product catalogs, health and life-science services, and more [11, 14, 25, 31]. Knowledge graphs are typically incomplete due to knowledge acquisition problems that happen during their creation, such as extraction errors, unreliable information and information disparity [8, 26]. Knowledge graphs comprise (subject, predicate, object) triples, where subject and object are entities, and predicate is the relationship's label.

Link prediction consists of training a machine learning model to predict missing triples [9]. Link prediction models typically output a score measuring the plausibility of a prediction that needs to be assessed against other scores [2, 7, 30, 32, 33, 39]. Assuming certain training, validation and test splits, the protocol to evaluate accuracy is as follows [9]: for each triple in the validation/test split, rank its score in ascending order with respect to the scores obtained when the triple's subject is replaced by all available entities. All these new triples are considered negatives. Then, obtain a similar rank but replacing the triple's object. A link prediction model is considered accurate if it outputs low ranks, i.e., if the positive triples are ranked higher than their negative counterparts. Note that this protocol corresponds to the transductive case in which all entities are known during training [3], which is our focus in this paper.

The evaluation protocol has two shortcomings: 1) It is computationally expensive since it requires to generate many negatives per positive, compute their scores, and compare them to determine ranks; and 2) Assessing the plausibility of a single triple is not possible, since it is mandatory to compare its score w.r.t. the scores of other triples derived from it. To address the first shortcoming, Bastos et al. [5] proposed to compute a distance between samples of positive and negative predictions made by a link prediction model. This distance is correlated to rank aggregations like mean rank. However, the distance provides a single value for the whole link prediction evaluation, and it does not take individual ranks into account; therefore, Bastos et al. [5] do not address the second shortcoming. To address the second one, two studies explored model calibration for knowledge graphs, which outputs the posterior probability (the probability to be plausible) for an input triple [34, 38]. Unfortunately, these two studies did not analyze link prediction evaluation, but triple classification and predicate prediction evaluation. Hence, these studies do not address the first shortcoming. As a result, to the best of our knowledge, there is no approach in the literature that has jointly addressed these two shortcomings. Such an approach is appealing since link prediction evaluation requires significant computational resources [5, 30], assessing individual triples is a must in production, and there is uncertainty about negatives due to the open-world assumption, i.e., it is unknown whether missing knowledge is correct or incorrect [26].

In this paper, we propose an alternative protocol for link prediction evaluation based on the posterior probabilities output by a calibration function. This function is learned during the validation step when training the link prediction model. We use the triple scores output by the link prediction model at hand, and align those

*TheWebConf'24, May 13–17, 2024, Singapore*
© 2024 Association for Computing Machinery.
ACM ISBN 978-x-xxxx-xxxx-x/YY/MM...$15.00
https://doi.org/10.1145/nnnnnnn.nnnnnnn

scores with expected positives and negatives. Instead of ranks, our alternative protocol considers only the posterior probabilities of the triples present in the test split (positives). Thus, generating negatives is no longer necessary, significantly reducing computation time (first shortcoming). The posterior probability of a triple $t_i$ determines its plausibility: $t_i$ is negative or positive if $f(x(t_i)) \in [0, 0.5)$ or $f(x(t_i)) \in [0.5, 1]$, respectively, where $f$ is the calibration function, and $x$ the function that assigns a plausibility score to $t_i$. Hence, we can determine the plausibility of $t_i$ without comparing it to other triples (second shortcoming).

How accurate and reliable are calibration functions for link prediction evaluation? Are the posterior probabilities output by these functions statistically correlated to ranks? What is the time reduction of the alternative protocol based on posterior probabilities? Can we rely on posterior probabilities to compare link prediction models side by side? We experimentally answer these questions using nine different link prediction methods that are diverse [1, 9, 16, 23, 35, 37, 41, 46, 47], i.e., they exploit a variety of mathematical constructs to compute scores, such as complex numbers, quaternions, and more. Calibration functions are learned using both Platt scaling and isotonic regression [24, 29], i.e., two different approaches to compute posterior probabilities. Our experimental results show that calibration functions achieve high accuracy and reliability. The posterior probabilities output by these functions exhibit high correlation with ranks used to evaluate link prediction. The time reduction of the original vs. our alternative protocols is significant, between 77% and 99%. When comparing models side by side, more than 78% of the individual comparisons are preserved between the original and our alternative protocols.

The summary of our contributions is as follows:

- We discuss how to learn a calibration function for link prediction evaluation using Platt scaling and isotonic regression. As far as we know, it is the first time this has been studied.
- We propose an alternative protocol for link prediction evaluation based on the output of the calibration function learned. This new protocol only works with positives.
- We propose several ways of assessing the accuracy and reliability of the calibration function learned, and of our alternative protocol to evaluate link prediction.
- We conduct experiments involving popular methods, such as BoxE, HAKE, QuatE and TransE, and datasets, such as FB15K-237, NELL-995, WN18RR and YAGO3-10.

The rest of the paper is organized as follows: Section 2 introduces link prediction evaluation and model calibration. Section 3 discusses our alternative protocol. Sections 4 and 5 respectively present how to learn and assess calibration functions. Section 6 describes our experiments and results. Section 7 presents the related work. Finally, Section 8 presents conclusions and future work.

## 2 Background

A knowledge graph $G$ is a set of $(s, p, o)$ triples, where $s$ and $o$ are entities and $p$ is a predicate. We use $E$ to denote the set of entities in $G$, which is partitioned into training, validation and test, i.e., $G_{TR}$, $G_{VA}$ and $G_{TE}$. A link prediction model comprises a number of embeddings (numerical vectors) that are associated to entities and predicates [43]. Each model exploits a scoring function $x(s, p, o)$

that assigns scores to input triples. These models are trained to minimize a loss function such that the model assigns low scores for positive triples (triples that belong to the graph), and high scores for negative triples (they do not belong to the graph).

Below, we introduce link prediction evaluation (Section 2.1) and model calibration (Section 2.2).

### 2.1 Link prediction evaluation

Currently, link prediction is evaluated using ranking-based metrics [9]. Since the goal is to predict low scores for positives, it follows that the metrics should measure a model's ability to do so. The idea is to register the position, rank, of a positive w.r.t. its negative counterparts when sorted by score in ascending order.

***Problem statement.*** Link prediction evaluation consists of, for each triple $t_i = (s, p, o) \in G_{TE}$, computing ranks $r_i^s$ and $r_i^o$ as follows:

$$r_i^e = 1 + |\{t_i' \mid t_i' \in N_{LW}^e(t_i, G) \wedge x(t_i) \leq x(t_i')\}| \qquad (1)$$

where $e$ can be either $s$ or $o$, and $x(t_i)$ is the scoring function of the model under evaluation applied over $t_i$.[1]

Above, $N_{LW}^e(t_i, G)$ computes the negative counterparts of $t_i$ considering $G$. Generating negative counterparts is challenging: knowledge graphs only contain positive triples [19], and they usually operate under the open-world assumption, i.e., triples that are not present in the graph at hand may be either missing or negatives [13]. Different strategies generate the negative counterparts of a positive triple $(s, p, o)$. The local-closed world assumption (LCWA) is the most popular one [9], which is as follows:

$$\begin{aligned} N_{LW}^s((s, p, o), G) &= \{(s', p, o) \mid (s', p, o) \notin G \wedge s' \in E\} \\ N_{LW}^o((s, p, o), G) &= \{(s, p, o') \mid (s, p, o') \notin G \wedge o' \in E\} \end{aligned} \qquad (2)$$

Intuitively, LCWA uses every entity present in $G$ as long as the corrupted triple is not in $G$. This is known as the filtered setting since the whole $G$ (no splits) is used to filter corrupted triples out [9]. The raw setting uses $G_{TE}$ only to filter corrupted triples out. Since the number of training and validation triples is typically larger than those in the test split, i.e., $|G_{TR} \cup G_{VA}| \gg |G_{TE}|$, the raw setting may consider many negatives that are present in $G_{TR} \cup G_{VA}$ and, therefore, are indeed positives. As a result, the filtered setting is preferred for link prediction evaluation [9].

***Metrics.*** We focus on $MR$, the mean of the ranks of the positive triples. $MR$ is recommended to evaluate link prediction since others like the mean reciprocal rank are problematic [17, 20, 39], e.g., when using reciprocal ranks, the difference between 1 and 2 is the same as between 2 and $\infty$.[2] $MR$ is computed as follows:

$$MR = \frac{1}{2|G_{TE}|} \sum_{t_i \in G_{TE}} r_i^s + r_i^o \qquad (3)$$

The $MR$ value of an accurate link prediction model is expected to be closer to one ($MR \in [1, \infty)$).

---

[1]In practice, we compute fractional ranks that differentiate between scores that are strictly less than $x(t_i)$ and tied to $x(t_i)$. Fractional ranks are preferred to mitigate the effect of ties in model scores [20].

[2]The difference is 0.5 in both cases since $\frac{1}{1} - \frac{1}{2} = \frac{1}{2} - \frac{1}{\infty} = 0.5$

## 2.2 Model calibration

Given a machine learning model, the goal is to learn a transformation function $f$ to produce a posterior probability [24], i.e., the probability of belonging to a class, positive or negative, based on the score output by the model. This is formalized as follows:

$$\hat{y}_i = f(x_i) \qquad (4)$$

where $\hat{y}_i$ is the posterior probability and $x_i$ is a score.

Platt scaling [29] defines a transformation function as follows:

$$\hat{y}_i = \sigma(a\,x_i + b) \qquad (5)$$

where $\sigma$ is the sigmoid function and $a$ and $b$ are two scalars that are learned using gradient descent. This implies that batches of scores can be provided to compute the calibration function.

Isotonic regression [24] is another method that uses a monotonically increasing transformation function $m$ as follows:

$$\hat{y}_i = m(x_i) \qquad (6)$$

Different from Platt scaling, function $m$ cannot be learned using gradient descent, which entails that all of the scores must be provided at once to compute the calibration function.

***Metrics.*** The weighted Brier score [45] is as follows:

$$BS_w = \frac{\sum_i w_i\,(\hat{y}_i - y_i)^2}{\sum_i w_i} \qquad (7)$$

where $w_i$ is the weight assigned to $\hat{y}_i$, and $y_i$ is the ground truth probability: one (zero) if triple $t_i$ is positive (negative). An accurate calibration function has a $BS_w$ value close to zero ($BS_w \in [0,1]$).

The weighted coefficient of determination [15] is as follows:

$$R_w^2 = 1 - \frac{\sum_i w_i\,(\hat{y}_i - y_i)^2}{\sum_i w_i\,(\bar{y} - y_i)^2} \qquad (8)$$

where $\bar{y}$ is the mean of the posterior probabilities $\hat{y}_i$. An accurate calibration function has an $R_w^2$ value close to one ($R_w^2 \in [0,1]$).

Using posterior probabilities, the 0.5 threshold separates positives and negatives. We compute true positives (*TP*), true negatives (*TN*), false positives (*FP*) and false negatives (*FN*) as follows:

$$TP = |\{\hat{y}_i \mid \hat{y}_i \geq 0.5 \wedge y_i = 1\}|; \; FP = |\{\hat{y}_i \mid \hat{y}_i \geq 0.5 \wedge y_i = 0\}|$$
$$TN = |\{\hat{y}_i \mid \hat{y}_i < 0.5 \wedge y_i = 0\}|; \; FN = |\{\hat{y}_i \mid \hat{y}_i < 0.5 \wedge y_i = 1\}| \qquad (9)$$

As discussed below, these values are aggregated to compute various measurements like true positive and negative rates.

## 3 Discussion

Focusing on Equation 2, we determine that the algorithmic complexity of link prediction evaluation is $O(|G_{TE}|\,|E|\,|G|)$. For every triple in $G_{TE}$, we corrupt either its subject or object using all entities ($E$), and check whether each corrupted triple is in $G$ (we assume linear search). The algorithmic complexity is simplified to $O(|E|)$ if we make the following assumptions: 1) Checking corrupted triples in $G$ is $O(1)$, e.g., there exists a hashing function; and 2) $|G_{TE}| \ll |E|$, which is common in practice. Iterating through all the entities to compute corrupted triples is thus the main component in the algorithmic complexity of link prediction evaluation.

We aim to find an alternative protocol for link prediction evaluation that reduces the $O(|E|)$ algorithmic complexity. Specifically, a desirable complexity is $O(|G_{TE}|)$ since it does not depend on the size of $G$ and, therefore, it is scalable. To accomplish our goal, we turn our attention to model calibration.

***Alternative protocol.*** Assume that there exists a calibration function $f$ defined over the scores output by a link prediction model. Link prediction evaluation consists of, for each triple $t_i = (s, p, o) \in G_{TE}$, computing $\hat{y}_i = f(x(t_i))$. A model's accuracy is measured using the mean of the posterior probabilities as follows:

$$\bar{y} = \frac{1}{|G_{TE}|} \sum_{t_i \in G_{TE}} \hat{y}_i \qquad (10)$$

***Roadmap.*** Our alternative protocol requires us to study the following: 1) How to learn a calibration function $f$ for a given link prediction model; and 2) How to assess reliability and accuracy of the alternative w.r.t. the original protocol based on ranks.

## 4 Learning $f$

As discussed by Platt [29], there are several considerations when learning a calibration function $f$. The main one is which split to use with the goal of reducing overfitting and bias. We also need to consider efficiency. Training a link prediction model is computationally expensive [30], so it is desirable that learning $f$ associated to the link prediction model has a low computational cost. The following observations are typically fulfilled in real-world datasets: 1) The training split is larger than the validation split, $|G_{TR}| \gg |G_{VA}|$. It is even common that $G_{TR}$ contains 85% or more of the total number of triples in $G$; and 2) The number of negatives synthetically generated is significantly larger than the number of positives. Commonly, there are close to $2|E|$ number of negatives per positive.

Because of these considerations and observations, there are several shortcomings when using $G_{TR}$ to learn $f$. First, the larger number of positives in $G_{TR}$ compared to $G_{VA}$ makes the learning of $f$ computationally expensive. This is exacerbated if we consider the negatives that are synthetically generated. Therefore, it is generally unfeasible to learn $f$ using the whole training split. Thus, positives and negatives must be sampled. As a result, the accuracy of the calibration functions heavily depends on said sampling process. Second, training both link prediction model and calibration function using the same protocol will lead to both reflecting the same biases. Hence, it is appealing to use a different split to learn $f$.

Using $G_{VA}$ is thus more appealing than using $G_{TR}$ to mitigate the previous shortcomings. Learning $f$ can therefore be accomplished during the training process of the link prediction model. The validation step evaluates early stopping criteria. Note that this step already computes scores for positives and negatives (as well as ranks) using $G_{VA}$. Learning $f$ consists of gathering these scores and fitting the function, without requiring extra resources.

Because of the second observation above, there are a significantly smaller number of positives in $G_{VA}$ than synthetically generated negatives. The training split of the calibration function is thus highly and intrinsically imbalanced. Tabacof and Costabello [38] proposed to sample a random subset of negatives per positive triple and apply a weighting scheme based on such selection. Since this

sampling can be biased depending on the negative counterparts selected, we propose to use the whole set of negatives, and the following weights for positives and negatives, respectively:

$$w_+ = \frac{1}{|G_{VA}|}; \qquad w_- = \frac{1}{|N_{LW}(G_{VA}, G_{TV})|} \qquad (11)$$

where $G_{TV} = G_{TR} \cup G_{VA}$ and $N_{LW}(G_{VA}, G_{TV})$ are the negatives generated by LCWA as follows:

$$N_{LW}(G_{VA}, G_{TV}) = \bigcup_{t \in G_{VA}} N_{LW}^s(t, G_{TV}) \cup N_{LW}^o(t, G_{TV}) \qquad (12)$$

Eq. 12 applies the filtered setting over the training and validation splits, i.e., the test split remains unseen since it will be used for evaluation purposes only.

***Expected behavior.*** Even though LCWA is the de facto standard strategy, we can use other strategies to identify negatives within LCWA. The main benefit is that these negatives have certain expected semantics and, therefore, they are useful to shed light on the behavior of the calibration function. Bansal et al. [4] defined the following strategies: global naïve (GB) and type-constrained LCWA (TC). These are extra conditions applied to LCWA. For a given triple $t = (s, p, o)$, the GB strategy is as follows [4]:

$$N_{GB}^s(t, G) = \{t' \mid t' = (s', p, o) \in N_{LW}^s(t, G) \wedge s' \notin S(G)\}$$
$$N_{GB}^o(t, G) = \{t' \mid t' = (s, p, o') \in N_{LW}^o(t, G) \wedge o' \notin O(G)\} \qquad (13)$$

where $S(G) = \{s \mid (s, p, o) \in G\}$ and $O(G) = \{o \mid (s, p, o) \in G\}$. Intuitively, $s'$ and $o'$ are entities that are never subjects and objects in $G$, respectively. For instance, assume a graph such that locations only appear as objects; negatives generated using GB contain locations as subjects, which is never the case in this graph. GB is expected to generate nonsensical negatives: entities that are never subjects (objects) are forced to be subjects (objects).

Similarly, the TC strategy is as follows [4]:

$$N_{TC}^s(t, G) = \{t' \mid t' = (s', p, o) \in N_{LW}^s(t, G) \wedge s' \in S(G, p)\}$$
$$N_{TC}^o(t, G) = \{t' \mid t' = (s, p, o') \in N_{LW}^o(t, G) \wedge o' \in O(G, p)\} \qquad (14)$$

where $S(G, p) = \{s \mid (s, p, o) \in G\}$ and $O(G, p) = \{o \mid (s, p, o) \in G\}$, i.e., $s'$ and $o'$ are entities that are subjects and objects in $G$ for the predicate $p$, respectively. Following the same example presented above, negatives generated using TC contain locations as objects, which is always the case in the graph at hand. TC is expected to generate negatives that are semantically plausible, i.e., they are more prone to be actual missing triples than other negatives.

We propose a third strategy, local naïve (LC), as follows:

$$N_{LC}^s(t, G) = \{t' \mid t' = (s', p, o) \in N_{LW}^s(t, G) \wedge s' \in C_{OS}(G, p)\}$$
$$N_{LC}^o(t, G) = \{t' \mid t' = (s, p, o') \in N_{LW}^o(t, G) \wedge o' \in C_{SO}(G, p)\} \qquad (15)$$

where $C_{OS}(G, p) = O(G, p) \setminus S(G, p)$ and $C_{SO}(G, p) = S(G, p) \setminus O(G, p)$. The expected semantics of these negatives is also nonsensical. However, different from GB, $s'$ is an entity that is object but never subject of predicate $p$ (similarly for $o'$). LC is more restrictive than GB: using the same example above, if a predicate does not

have any locations as objects, these locations are never used to corrupt subjects, while these locations appear when applying GB.

## 5 Assessing $f$

Once $f$ has been learned, we aim to evaluate its reliability and accuracy. The weighted Brier score ($BS_w$) and the weighted coefficient of determination ($R_w^2$) presented above are proper assessment measurements. We use the test split, $G_{TE}$, for evaluation purposes, which has the same drawback as the validation split: it is highly imbalanced. We propose to use the same weighting scheme as define in Equation 11, where we consider the total number of negatives for a given strategy that synthetically generates negatives.

We also compute true and false positives and negatives as described above (Equation 9), and we aggregate them to compute different measurements like precision and recall. However, certain aggregations are more appealing than others. The reason why we rely on negative generation strategies is because knowledge graphs typically operate under the open-world assumption. This entails that positive triples in the test split are generally true.[3] However, we are uncertain whether generated negative triples are indeed negatives. We thus expect *TP* and *TN* to be large and *FN* to be small. However, we cannot make any assumptions about *FP*: some negatives may be considered positives because they are indeed positives. To mitigate this issue, we propose to use aggregations that consider positives and negatives independently. We focus on true positive and negative rates as follows:

$$TPR = \frac{TP}{TP + FN}; \quad TNR = \frac{TN}{TN + FP} \qquad (16)$$

Note that other aggregations like precision will yield poor results because of the imbalanced nature of the datasets. This is the case for every aggregation that combines *TP* and *FP*, or *TN* and *FN*, which have different upper bounds. Both *TPR* and *TNR* can be combined into a single balanced accuracy measurement as follows:

$$BA = \frac{TPR + TNR}{2} \qquad (17)$$

***Expected behavior.*** Even though LCWA is the most common strategy, other negative generation strategies are also appealing to assess calibration functions. Specifically, using the expected semantics of these strategies is useful to gain additional insights on these functions. Since GB and LC are expected to generate nonsensical negatives, calibration functions should easily discern between positives and negatives. $BS_w$ and $BA$ values when using GB and LC should thus remain low and high, respectively. Similarly, calibration functions should not be as accurate when using the TC strategy, since negatives are expected to be semantically plausible. $BS_w$ and $BA$ values should thus increase and decrease, respectively.

***Comparison with ranks.*** We wish to assess our alternative and the original protocols for link prediction evaluation. To do so, we propose to compare the relationship between posterior probabilities output by $f$, and the ranks computed during link prediction evaluation, i.e., $r^s$ and $r^o$ for each triple in the test split. The ideal scenario is that, for a given pair of link prediction model and calibration

---

[3]Some of the triples may be incorrect due to acquisition errors. We assume that, if any, the number of incorrect triples is small, so we ignore the presence of errors.

function, ranks are linearly correlated with posterior probabilities. We thus focus on the Pearson correlation coefficient as follows:

$$r_{xy} = \frac{\sum_i (x_i - \bar{x})(\hat{y}_i - \bar{y})}{\sqrt{\sum_i (x_i - \bar{x})^2}\sqrt{\sum_i (\hat{y}_i - \bar{y})^2}} \qquad (18)$$

where $x_i$ is either $r^s$ or $r^o$ for a given triple, and $\bar{x}$ is the mean rank, i.e., *MR*. Correlation values range between -1 and 1, such that high correlation implies that $r_{xy} \simeq -1$ or $r_{xy} \simeq 1$, while $r_{xy} = 0$ means no correlation. There are two considerations. On one hand, ranks have different upper bounds [39]: for a triple $t$ with ranks $r^s$ and $r^o$, $r^s$ and $r^o$ are respectively upper-bounded by $|N^s_{LW}(t, G)|$ and $|N^o_{LW}(t, G)|$. Therefore, we adjust ranks as follows:

$$\breve{r}^s = 1 - \frac{r^s - 1}{|N^s_{LW}(t, G)|}; \quad \breve{r}^o = 1 - \frac{r^o - 1}{|N^o_{LW}(t, G)|} \qquad (19)$$

On the other hand, a single triple has two relative ranks associated to it, $\breve{r}^s$ or $\breve{r}^o$, but a single posterior probability, $\hat{y}$. We duplicate posterior probabilities and produce the following two pairs for each triple: $(\breve{r}^s, \hat{y})$ and $(\breve{r}^o, \hat{y})$, which we use to compute $r_{xy}$.

## 6 Experiments

In this section, we discuss the experiments we conducted. We present the datasets we used, and how we trained and learned link prediction models and calibration functions (Section 6.1). We discuss accuracy and reliability results (Section 6.2), as well as time results and comparisons between protocols (Section 6.3).

### 6.1 Datasets and models

The datasets we used in our experiments and their total number of entities ($|E|$), predicates ($|R|$) and triples ($|T|$) are as follows:

|               | $|E|$   | $|R|$ | $|T|$     |
|---------------|---------|-------|-----------|
| BioKG [42]    | 105,524 | 17    | 2,067,998 |
| FB15K [9]     | 14,951  | 1,345 | 592,213   |
| FB15K-237 [40]| 14,541  | 237   | 310,116   |
| Hetionet [18] | 45,158  | 24    | 2,250,197 |
| NELL-995 [44] | 75,492  | 200   | 139,874   |
| WN18 [9]      | 40,943  | 18    | 151,442   |
| WN18RR [12]   | 40,943  | 11    | 93,003    |
| YAGO3-10 [36] | 123,182 | 37    | 1,089,040 |

These are commonly used to evaluate link prediction [2, 7, 30, 32, 33, 39], are publicly available, and already partitioned into training, validation and test splits. BioKG integrates a number of biological databases into a single knowledge graph. FB15K and FB15K-237 were extracted from Freebase [6], Hetionet from the Hetionet integrative network constructed using millions of biomedical studies [18], NELL-995 from the Never-Ending Language Learning project [28], WN18 and WN18RR from WordNet [22], and YAGO3-10 from YAGO [21]. Note that FB15K-237 and WN18RR are subsets of FB15K and WN18, respectively, in which redundancy has been reduced. Furthermore, the validation and test splits publicly available for NELL-995 contain 0.4% and 2.6% of the triples, respectively. However, we found that the size of the validation split and the variety

of the triples contained in it were detrimental to learn calibration functions. Therefore, we decided to reconfigure these validation and test splits to contain 1.6% of the triples each.

We trained a number of link prediction models [4] using the following link prediction methods: BoxE [1], ComplEx [41], HAKE [47], HolE [23], QuatE [46], RotatE [37], RotPro [35], TorusE [16], and TransE [9]. These methods are diverse and use a variety of approaches: rectangles in the Euclidean space, complex numbers, points in a polar coordinate system, circular correlations in the Euclidean space, quaternion and rotations in the quaternion space, rotations and projection functions in the complex space, toruses in the Euclidean space, and Euclidean or Manhattan distances over real numbers. (See Appendix A for training details.)

For each link prediction model, we learned a single calibration function using isotonic regression, and six additional functions using Platt scaling, where we varied parameter initialization as follows: $\{a, b\} \subseteq \{-1, 0, 1\}$. We used the validation split to learn $f$, and LCWA to generate negatives as described above.

### 6.2 Accuracy and reliability results

What is the reliability and accuracy of the best calibration functions? Which calibration approach is superior, Platt scaling or isotonic regression? Are posterior probabilities correlated to relative ranks? Is the behavior of the calibration functions as expected when using different negative generation strategies?

We used $R^2_w$ over the validation split to determine the best calibration function for each link prediction model. Figure 1 presents the five-number summary (min, max, mean, and first and third quartiles) of the accuracy of these models over the test split. We aggregate all of the $R^2_w$ values for the datasets and group by link prediction method. We observe that all mean values are between 0.7 and 0.9 (see Figure 1a). This indicates that the posterior probabilities output by the calibration functions are good approximations of the ground truth of positive and synthetically-generated negatives. Similarly, the $BS_w$ mean values are less or equal than 0.1 (see Figure 1b), which entails that posterior probabilities are properly calibrated. These best calibration functions also exhibit high *BA* values with mean values greater or equal than 0.9 (see Figure 1c). They are thus able to accurately discern between positives and negatives using the 0.5 threshold over the posterior probabilities. We also studied the correlation between relative ranks and posterior probabilities (see Figure 1d). Even though the $r_{xy}$ mean values are greater than 0.55, we observe differences among calibration functions. The functions for the ComplEx, HolE and QuatE models are the ones achieving the best and most consistent correlation results, with mean $r_{xy}$ values close to 0.8. In the next group, HAKE and TorusE achieve mean values closer to 0.6. RotPro and TransE are the next ones, with RotPro exhibiting a larger min value. Finally, BoxE and RotatE exhibit different behavior: the former achieves consistent correlation values around 0.6, while the range of values for RotatE is uneven between 0.4 and 0.8 with a mean of 0.7 (approximately). We note that the BoxE and RotatE link prediction models achieve uneven accuracy using ranks (described below), which is the reason why these correlation results deviate.

---

[4]Models, source code and results are publicly available. We will disclose the URL after the double-blind review process.

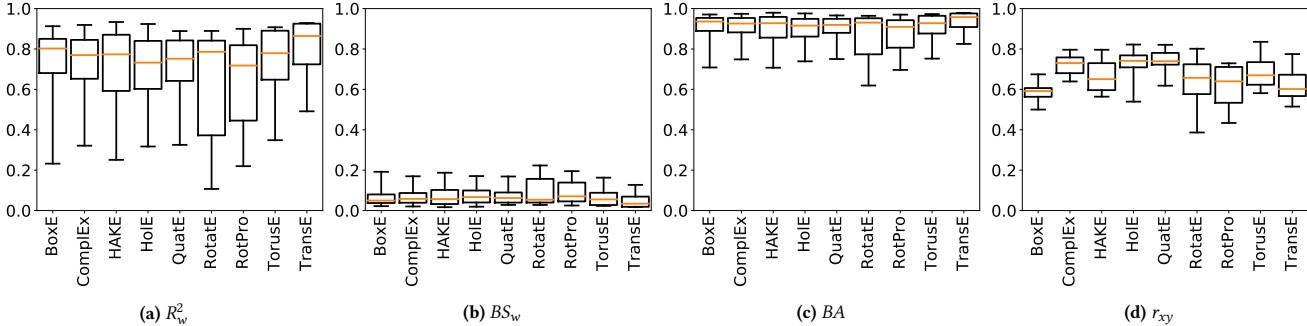

**Figure 1: Five-number summary assessing the best calibration functions using LCWA to generate negatives. We aggregate the results obtained over the test split of each dataset under evaluation and group by link prediction method.**

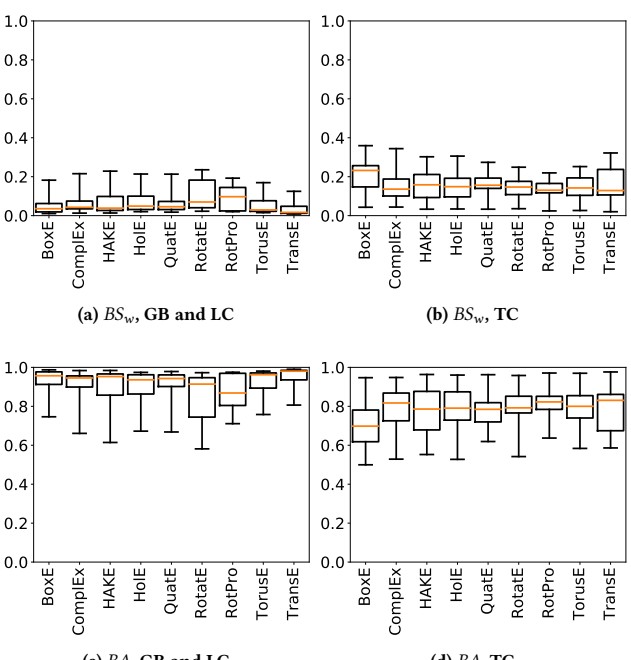

**Figure 2: Five-number summary assessing the best calibration functions using the GB and LC (nonsensical) and the TC (semantically plausible) strategies, respectively. Results are aggregated and grouped by link prediction method.**

To study expected behavior, we combine the GB and LC strategies presented above, which are expected to output nonsensical negatives only. We also study TC that is expected to output semantically plausible negatives only. Figure 2 compares the results for both settings side by side. All the calibration functions behave as expected: the functions are able to better discern between nonsensical negatives and positives than between semantically-plausible negatives and positives. BoxE and TransE achieve best and second-best consistent behavior. Using these results, we can better understand some of previous results. For instance, we observe that RotatE and

RotPro achieve a wider range of $BS_w$ and $BA$ values. Specifically, RotPro exhibits better values discerning negatives generated using TC than GB and LC, which is unappealing.

***Takeaways****.* Calibration functions generally achieve competitive accuracy measured using $R^2_w$, $BS_w$ and $BA$. Without exception, all of the calibration functions that achieve the best $R^2_w$ results over the validation split were learned using isotonic regression. In other words, none of the functions learned using Platt scaling achieved better results than those using isotonic regression. Furthermore, many calibration functions exhibit high correlation ($r_{xy}$) between posterior probabilities and relative ranks. The exceptions are BoxE, RotatE and RotPro, which achieve the worse correlation results and, at the same time, exhibit uneven accuracy results measured using ranks. Calibration functions generally behave as expected when different strategies to generate negatives are exploited.

### 6.3 Time and comparison results

Does the time taken to compute link prediction using our alternative protocol improve w.r.t. the original protocol? For the best-performing calibration functions, we study the time reduction between both protocols. Specifically, $t_A$ is the time taken to learn $f$ over $G_{VA}$ plus the time taken to compute posterior probabilities over $G_{TE}$. The time difference is $\%\Delta t = 100\,(t_A + t_{LP})/t_{LP}$, where $t_{LP}$ is the time taken to evaluate link prediction using the original protocol. In Table 1, the time reduction is significant, more than 90%, in many cases. We also observe that RotPro and TorusE are generally the most and the least benefited from the alternative protocol, respectively. However, there is still a 77% time reduction for the TorusE model over NELL-995, which is the overall minimum.

The next question we aim to answer is: can we compare models side by side based on posterior probabilities instead of ranks? Even though there may not be a strong linear correlation ($r_{xy}$) between ranks and posterior probabilities, this correlation considers both positives and negatives, while our alternative protocol focuses on positives only. To answer the question, we sort link prediction models and calibration functions by accuracy. For presentation purposes, we scale $MR$ as follows: $MR' = 1 - (MR - min(MR))/(max(MR) - min(MR))$, where $min(MR)$ and $max(MR)$ are the minimum and maximum $MR$ values obtained for all the models over the dataset

**Table 1: Time in seconds taken to evaluate link prediction using our alternative protocol ($t_A$) and time difference between original vs. alternative protocols ($\%\Delta t$). Best and worst results are in bold and underlined, respectively.**

| | BioKG | | FB15K | | FB15K-237 | | Hetionet | | NELL-995 | | WN18 | | WN18RR | | YAGO3-10 | |
|---|---|---|---|---|---|---|---|---|---|---|---|---|---|---|---|---|
| | $t_A$ | $\%\Delta t$ | $t_A$ | $\%\Delta t$ | $t_A$ | $\%\Delta t$ | $t_A$ | $\%\Delta t$ | $t_A$ | $\%\Delta t$ | $t_A$ | $\%\Delta t$ | $t_A$ | $\%\Delta t$ | $t_A$ | $\%\Delta t$ |
| BoxE | 533.7 | -92.8 | 766.4 | -98.4 | 352.5 | -97.3 | 191.5 | -92.6 | 153.2 | -90.6 | 192.7 | -92.5 | 113.8 | -91.4 | 587.8 | -91.8 |
| ComplEx | 544.4 | -88.2 | 752.8 | -98.7 | 230.0 | -96.4 | 195.1 | -87.9 | 154.9 | -82.7 | 195.0 | -87.6 | 116.9 | -84.8 | 608.2 | -87.5 |
| HAKE | 610.4 | -89.1 | 926.8 | -98.5 | 273.4 | -96.6 | 200.5 | -87.8 | 180.8 | -81.2 | 240.8 | -96.1 | 114.3 | -86.1 | 644.4 | -87.4 |
| HolE | 488.0 | -85.3 | 732.3 | **-98.9** | 229.6 | -96.8 | 192.1 | -87.0 | 191.8 | -80.8 | 186.3 | -87.2 | 109.1 | -83.0 | 578.1 | -85.2 |
| QuatE | 538.5 | -91.9 | 837.7 | -98.3 | 220.8 | -96.6 | 208.5 | -91.5 | 188.0 | -89.7 | 227.0 | -92.3 | 128.4 | -90.8 | 623.8 | -91.6 |
| RotatE | 536.0 | -90.1 | 838.9 | -98.7 | 240.1 | -96.4 | 199.3 | -89.2 | 165.1 | **-95.2** | 183.2 | -91.2 | 112.1 | -87.4 | 601.5 | -89.1 |
| RotPro | 479.6 | -91.9 | 933.2 | -98.7 | 282.2 | -97.1 | 201.9 | **-93.9** | 141.2 | -91.1 | 262.3 | **-96.2** | 142.7 | **-96.9** | 638.0 | **-92.6** |
| TorusE | 673.2 | -92.9 | 777.4 | -98.7 | 300.5 | **-98.3** | 199.2 | -87.0 | 157.1 | -77.0 | 177.8 | -86.2 | 112.9 | -82.0 | 613.7 | -87.1 |
| TransE | 553.1 | **-93.4** | 995.4 | -98.6 | 286.3 | -97.1 | 194.6 | -92.6 | 179.4 | -89.9 | 285.2 | -93.7 | 151.4 | -95.1 | 567.1 | -85.6 |

**Table 2: Link prediction models and calibration functions sorted using scaled $MR$ ($MR'$) and mean posterior probability of positives ($\bar{y}$). We also present the overlap between the individual comparisons of both orders. Names are abbreviated as follows: Bo=BoxE, Co=ComplEx, HA=HAKE, Ho=HolE, Qu=QuatE, Rt=RotatE, RP=RotPro, To=TorusE, Tr=TransE.**

| Dataset | | Sorted by accuracy | Overlap |
|---|---|---|---|
| BioKG | $MR'$ | Tr (1.00), HA (1.00), Co (0.94), Ho (0.94), Qu (0.90), To (0.86), Bo (0.86), Rt (0.51), RP (0.00) | |
| | $\bar{y}$ | HA (0.97), Tr (0.96), Ho (0.96), Co (0.96), To (0.95), Qu (0.94), Bo (0.94), Rt (0.93), RP (0.89) | 92% |
| FB15K | $MR'$ | Tr (1.00), Bo (1.00), To (0.69), HA (0.66), Rt (0.65), RP (0.52), Co (0.42), Qu (0.30), Ho (0.00) | |
| | $\bar{y}$ | Tr (0.96), Bo (0.96), Rt (0.94), To (0.94), RP (0.94), HA (0.93), Co (0.93), Qu (0.91), Ho (0.91) | 92% |
| FB15K-237 | $MR'$ | Tr (1.00), Bo (0.96), To (0.73), Rt (0.67), HA (0.63), Qu (0.57), Co (0.42), Ho (0.20), 'RP (0.00) | |
| | $\bar{y}$ | Tr (0.93), Bo (0.92), Rt (0.89), To (0.89), Qu (0.88), Co (0.87), Ho (0.84), HA (0.84), RP (0.84) | 89% |
| Hetionet | $MR'$ | HA (1.00), RP (0.94), Bo (0.87), Rt (0.85), Tr (0.59), To (0.46), Co (0.28), Ho (0.02), 'Qu (0.00) | |
| | $\bar{y}$ | RP (0.90), Rt (0.89), HA (0.89), To (0.89), Ho (0.88), Bo (0.88), Tr (0.88), Co (0.88), Qu (0.87) | 78% |
| NELL-995 | $MR'$ | Bo (1.00), Tr (0.83), To (0.72), Co (0.71), Qu (0.58), HA (0.54), Ho (0.49), RP (0.37), 'Rt (0.00) | |
| | $\bar{y}$ | Bo (0.78), Tr (0.75), Qu (0.71), Co (0.71), To (0.67), Ho (0.66), HA (0.65), RP (0.61), Rt (0.55) | 89% |
| WN18 | $MR'$ | Tr (1.00), To (0.91), RP (0.64), Bo (0.37), Ho (0.35), HA (0.23), Rt (0.23), Qu (0.19), Co (0.00) | |
| | $\bar{y}$ | Tr (0.96), To (0.95), RP (0.95), Qu (0.94), Ho (0.94), HA (0.94), Co (0.92), Bo (0.92), Rt (0.92) | 78% |
| WN18RR | $MR'$ | Tr (1.00), To (0.79), RP (0.75), Rt (0.56), Bo (0.32), Ho (0.26), Co (0.20), Qu (0.19), 'HA (0.00) | |
| | $\bar{y}$ | Tr (0.80), To (0.74), RP (0.73), Rt (0.70), Ho (0.67), Qu (0.67), Co (0.66), HA (0.63), Bo (0.62) | 86% |
| YAGO3-10 | $MR'$ | Tr (1.00), Bo (0.96), HA (0.90), Co (0.90), To (0.86), Qu (0.85), Ho (0.78), RP (0.26), 'Rt (0.00) | |
| | $\bar{y}$ | Tr (0.93), Co (0.89), HA (0.88), Bo (0.87), Qu (0.86), To (0.85), Ho (0.85), RP (0.71), Rt (0.65) | 89% |

at hand. The best link prediction model achieves $MR' = 1$, and the worst performing model achieves $MR' = 0$. Let $L = \langle l_0, l_1, \ldots, l_8 \rangle$ be a sequence of link prediction models or calibration functions $l_i$ sorted by accuracy (either $MR'$ or $\bar{y}$), and $L' = \{l_0 > l_1, l_0 > l_2, \ldots\}$ be the set of individual comparisons in $L$. For each dataset, we compute the overlap between $L'_{MR'}$ and $L'_{\bar{y}}$, i.e., the number of individual comparisons that are shared when we sort by accuracy using $MR'$ and $\bar{y}$, respectively.

In Table 2, we observe that overlap values greater or equal than 78%. The order between models and functions was computed without rounding; there are no ties among the non-rounded values. In many cases, we observe that the $\bar{y}$ values are very close and, therefore, the order is altered. For instance, in Hetionet and WN18, the worst cases, all these values are between 0.90 and 0.87, and 0.96 and 0.92, respectively. These close values are the main reason why the overlap is only 78%. In the other datasets where $\bar{y}$ values are not so close, the overlap increases to 86% or more.

*Takeaways.* With up to -98.9%, the time reduction of our alternative protocol is significant. The mean of the posterior probabilities using positives can be indeed used to compare models side by side in a given dataset instead of ranks.

## 7 Related work

We discuss link prediction evaluation (Section 7.1), and model calibration for knowledge graphs (Section 7.2).

### 7.1 Link prediction evaluation

Bastos et al. [5] is the most related approach. Given a link prediction model, they construct two subgraphs by sampling positive and negative triples, respectively. Each triple in these new subgraphs are labeled with the scores assigned by the link prediction model under evaluation. The distance between both subgraphs is computed using persistent homology, a method that serves to compare topological features. This distance is correlated with rank aggregations like mean rank and mean reciprocal rank. This approach outputs a

single, global value for each link prediction model (the distance between the sampled positive and negative subgraphs). In our case, we deal with individual predictions made by link prediction models, which entails that our approach has a finer level of granularity. Additionally, calibration functions can be used in production to assign posterior probabilities to new, unseen triples, and decide whether they are positives or negatives without comparing to other triples. This is not possible with the approach by Bastos et al. [5].

Pezeshkpour et al. [27] conducted an experiment in which triple scores from link prediction models were directly transformed into posterior probabilities applying the sigmoid function, i.e., $\sigma(x(t_i))$ for a given triple $t_i$. The experiment compared positive triples with a sample of negative triples that were selected using three different strategies: LCWA, TC and another strategy similar to LC that takes types into account. The authors found that link prediction models are generally overconfident. The experiment did not consider Platt scaling or isotonic regression to compute posterior probabilities. The experiment relied on negative sampling, which is unappealing since different results may be observed for different samples. Applying the sigmoid function directly to model scores can be detrimental. The sigmoid function assumes that input values lie in the range of $(-\infty, \infty)$. However, distance-based methods like RotatE and TransE produce scores in the $(0, \infty)$ range, which implies that all these posterior probabilities will lie in the $(0.5, 1]$ range, i.e., they are all considered positives. Chen et al. [10] evaluated two types of functions to compute posterior probabilities: $\sigma(a\,x(t_i) + b)$, equivalent to Platt scaling, and $min(\,max(\,a\,x(t_i) + b, 0), 1)$. Their proposal aims to enable link prediction for uncertain knowledge graphs rather than link prediction evaluation.

There have been many studies in the context of link prediction evaluation [2, 7, 30, 32, 33, 39]. All of them focus on different aspects of the link prediction evaluation protocol, e.g., how hyperparameter optimization, graph splits and/or loss functions affect reported accuracy results. These studies are complementary to our work: model calibration can be applied to link prediction models learned in various ways.

## 7.2 Model calibration for knowledge graphs

Tabacof and Costabello [38] focused on triple classification evaluation. Different from link prediction, triple classification is a binary classification task that places every triple into one of two classes: positives that belong to the graph or negatives that do not. Tabacof and Costabello [38] applied model calibration to several triple classification models in which the calibration function was learned by sampling synthetically-generated negatives instead of those available in the ground truth. These learned calibration functions achieved comparable results to those learned using negatives in the ground truth. We focus on link prediction evaluation in which there is no ground truth of negatives available. Furthermore, sampling synthetically-generated negatives may significantly alter the output of the calibration function and, therefore, the learning is not deterministic. In our case, we use the whole set of synthetically-generated negatives, so the learning process of our calibration functions is deterministic.

Safavi et al. [34] studied the difference in the effectiveness of model calibration between open-world and closed-world assumptions for predicate prediction. Different from link prediction, predicate prediction aims to determine the missing predicate between two entities, subject and object. They used multiclass model calibration, in which each class is a given predicate in the knowledge graph at hand. The authors showed that model calibration for the closed-world assumption results in low expected calibration errors and high accuracy. We focus on link prediction rather than predicate prediction, which is more challenging since the number of predicates in knowledge graphs is generally orders of magnitude smaller than the number of entities. For the same reason, multiclass model calibration for link prediction is unfeasible.

## 8 Conclusions

There are a large number of link prediction methods that aim to predict missing triples. These methods are evaluated following a well-defined protocol: for every positive triple, generate a number of negative counterparts by corrupting its subject and object, but not both at the same time. Positives and negatives are sorted by score and the ranks of the positives are recorded. Accuracy is measured based on rank aggregations and it is expected that these ranks are close to one. Evaluating accuracy is thus a challenging task. Reducing the computational cost of this task is appealing; however, only a single approach has focused on such a reduction, as far as we know. The reduction consists of computing a single measurement for the whole evaluation that is correlated to rank aggregations. Unfortunately, this does not solve other shortcomings like determining the plausibility of an individual triple in isolation.

We explore the use of model calibration for efficient link prediction evaluation. A calibration function transforms link prediction scores into posterior probabilities such that 0.5 is the threshold to distinguish between positives and negatives. In our approach, first, each posterior probability indicates the plausibility of an individual triple. Second, learning calibration functions is accomplished at the validation step during the training of a link prediction model. Evaluation consists of computing the mean of the posterior probabilities of the positive triples, so negative counterparts can be avoided. Third, different strategies to generate negatives can be exploited when training and evaluating calibration functions. Our experiments show that the computational cost of link prediction evaluation is significantly reduced, that there is a high correlation between ranks and posterior probabilities, and that model comparisons using the mean of the posterior probabilities is similar to comparisons using mean ranks.

As future work, we aim to train a variety of link prediction models by altering the strategies to generate negatives, loss functions and graph splits. We will learn calibration functions for these link prediction models, and we will study their posterior probabilities. Besides being more efficient, we hypothesize that posterior probabilities will help shed more light on model behavior than ranks. We aim to apply posterior probabilities to interpret link prediction evaluation.

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

# A Training details

Link prediction methods have a number of hyperparameters that must be fine-tuned. For each method, we used a Sobol sequence to generate quasi-random, low-discrepancy combinations of hyperparameter values with the goal of evenly cover the space formed by these hyperparameters [39]. We evaluated the accuracy of each model after ten epochs using *MR* over the validation split. These *MR* values were provided to a Bayesian optimizer that suggested new combinations of values to explore, which were evaluated in the same fashion. We selected the best model after exploring twenty combinations. We relied on the Ax platform.[5] We used LCWA to generate negatives and sampled a number of them to form batches during training; therefore, the link prediction models were trained using stochastic gradient descent in mini-batch mode [9].

# B Platt scaling

Without exception, all the best performing calibration functions use isotonic regression. The question is: what is the accuracy of Platt scaling?

---

[5]https://ax.dev/

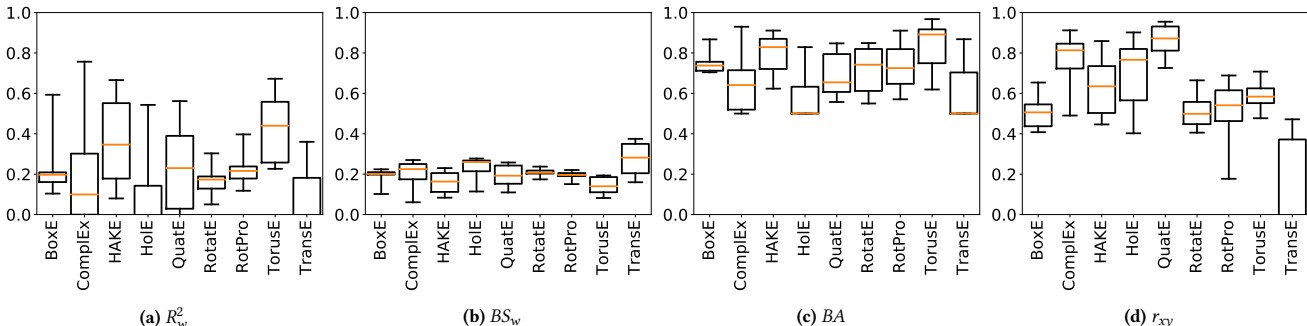

**(a)** $R_w^2$ **(b)** $BS_w$ **(c)** $BA$ **(d)** $r_{xy}$

**Figure 3: Five-number summary of the accuracy of the calibration functions learned using Platt scaling and LCWA to generate negatives. Results are aggregated and grouped by link prediction method.**

Figure 3 presents the accuracy results for these calibration functions. The $R_w^2$ and $BS_w$ values are significantly lower and higher, respectively, than those achieved by isotonic regression. It is notable that the mean of the $R_w^2$ values is negative in the cases of HolE and TransE. These results indicate that the learned functions are not properly calibrated. Similarly, all $BA$ values are also lower than those achieved by the functions learned using isotonic regression. However, some functions achieve reasonable accuracy with a mean greater than 0.75: BoxE, HAKE, RotatE, RotPro and TorusE.

Correlation results for these functions are uneven. Many calibration functions using Platt scaling achieve lower $r_{xy}$ values than their isotonic regression counterparts, e.g., RotPro. We also observe that the calibration functions for the TransE models have a wide range of values, which was not the case for isotonic regression.

Finally, the functions for the ComplEx models also have a wider range of values than using isotonic regression, and some of these values slightly outperform the previous results. This is more evident in the QuatE models, which achieve better correlation results than the isotonic regression functions.

We conclude that functions learned using Platt scaling are generally not well calibrated. Even though, a few of them achieve better correlation results than isotonic regression, it is unappealing to rely on poor calibrated functions. Isotonic regression is thus preferred in the context of link prediction evaluation. Also, learning $f$ using Platt scaling is more efficient than using isotonic regression. However, the performance of isotonic regression is generally quite superior.