# OpenReview forum: "Using Model Calibration to Evaluate Link Prediction in Knowledge Graphs"
_ACM.org/TheWebConf/2024/Conference — TheWebConf24_

### Official Review · Reviewer_FvPt · 2023-11-06

**Novelty:** 5
**Technical Quality:** 5

**Review:**

In the article “Using Model Calibration to Evaluate Link Prediction in Knowledge Graphs” authors propose to make a contribution to the evaluation of methods that aim to predict missing triples in a triplestore.

The main problem with that article is that its contribution is purely about KR / KG and no link to the Web is made.
So it violates the relevance rule: "Every submission must clearly state how the work is relevant to the Web and to the track in the first page. Submissions that merely use a Web artifact---e.g., a dataset or a Web Application Programmer Interface (API) or a social network---rather than answering a specific Web-related scientific research challenge, are out of scope and will be desk-rejected."

This paper never formulated any explicitly Web-related scientific research challenge.

It is clear from the lists of contributions page 2 that the submission would have been suitable for a purely AI/KG conference but not for TheWebConf:
“• We discuss how to learn a calibration function for link prediction evaluation using Platt scaling and isotonic regression. As far as we know, it is the first time this has been studied.
• We propose an alternative protocol for link prediction evaluation based on the output of the calibration function learned. This new protocol only works with positives.
• We propose several ways of assessing the accuracy and reliability of the calibration function learned, and of our alternative protocol to evaluate link prediction.
• We conduct experiments involving popular methods, such as BoxE, HAKE, QuatE and TransE, and datasets, such as FB15K-237, NELL-995, WN18RR and YAGO3-10.”

**Questions:**

How is you paper compliant with the relevance rule of the CfP of TheWebConf?
"Every submission must clearly state how the work is relevant to the Web and to the track in the first page. Submissions that merely use a Web artifact---e.g., a dataset or a Web Application Programmer Interface (API) or a social network---rather than answering a specific Web-related scientific research challenge, are out of scope and will be desk-rejected."
What is your paper contributing to the Web?

**Reviewer Confidence:**

2: The reviewer is willing to defend the evaluation, but it is likely that the reviewer did not understand parts of the paper

**Scope:**

3: The work is somewhat relevant to the Web and to the track, and is of narrow interest to a sub-community

---

### Official Review · Reviewer_b9uU · 2023-11-17

**Novelty:** 4
**Technical Quality:** 3

**Review:**

This paper studies the efficiency issues of evaluating KG link prediction models. It is well-written and easy to follow. The proposed method using model calibration is intuitive and straightforward. My detailed comments are as follows:

1. Although the efficiency problem of evaluating link prediction models seems to be important according to the complexity analysis in section 3, it is worth noting that in practice this task is performed in parallelized on GPUs. This makes the motivation of this paper less convincing. More importantly, it is unclear whether this consideration is taken into account in the experiments. What is the evaluation setting and hardware for the results in Section 6.3?

2. The proposed calibration method seems to be specific to ranking-based scoring functions (with negative samples). How about the other case of using cross-entropy loss (learning without negative samples), such as ConvE?

3. The application of classical calibration methods (Platt scaling and Isotonic regression) is intuitive, but lacks novelty. In particular, the experiments show that there are unexpected results on some link prediction techniques BoxE, RotatE, and RotPro in Section 6.2 and also on some datasets Hetionet and WN18 in Section 6.3, without a strong reason for explanation. Further investigation on these issues and designing a corresponding calibration method robust against these cases would be a strong plus for this work.

4. One more suggestion is to also consider instance completion tasks in KGs, such as prediction (h,?,?), see the reference below. The efficiency issue is more serious in this task.
- Rosso, Paolo, Dingqi Yang, Natalia Ostapuk, and Philippe Cudré-Mauroux. "Reta: A schema-aware, end-to-end solution for instance completion in knowledge graphs." In Proceedings of the Web Conference 2021, pp. 845-856. 2021.

**Questions:**

See the comments 1, 2, and 3 above.

**Reviewer Confidence:**

4: The reviewer is certain that the evaluation is correct and very familiar with the relevant literature

**Scope:**

4: The work is relevant to the Web and to the track, and is of broad interest to the community

---

### Official Review · Reviewer_Y3XG · 2023-11-23

**Novelty:** 5
**Technical Quality:** 5

**Review:**

This article introduces a novel protocol that leverages posterior probabilities of positive outcomes, rather than ranking systems, to evaluate link predictions in knowledge graphs. It also details a calibration function designed to assign posterior probabilities to edges.

The paper is well-composed and presents its ideas clearly. The subject matter is pertinent to the field, and the analysis of the new technique, which focuses on posterior probabilities, effectively highlights the limitations of current methodologies. The approach appears sound and is elaborately described.

The evaluation of this methodology is thorough, applying it to nine alternative methods across eight well-established benchmarks in the field. Notably, this approach significantly reduces the time required for computing link prediction.

However, the paper states that "Models, source code, and results will be made publicly available, with the URL to be disclosed after the double-blind review process." This presents a challenge for reviewers, as they are unable to examine these materials during the review process. In today's context, it is relatively straightforward to anonymously share such materials for conference review, and the lack of this provision is a notable shortcoming of the paper.

**Questions:**

Can you share the material on an anonymous link?

**Ethics Review Description:**

no concerns

**Reviewer Confidence:**

2: The reviewer is willing to defend the evaluation, but it is likely that the reviewer did not understand parts of the paper

**Scope:**

4: The work is relevant to the Web and to the track, and is of broad interest to the community

---

### Official Review · Reviewer_Yaad · 2023-11-27

**Novelty:** 6
**Technical Quality:** 7

**Review:**

This work proposes a protocol based on posterior probabilities to assess link prediction models in knowledge graphs instead of using ranks. This protocol consists of learning a posterior probability function $f$ in the evaluation step. Given a score function $x$ for triples $t$, the posterior probability is $f(x(t))$. The authors evaluate the new protocol on various embedding models, and datasets. The results show that their protocol has significant improvements regarding the time required for model evaluation.

The proposed method would have a high impact since the issues of the protocol based on ranks.

The paper is very well-written and sound.

**Questions:**

1. In Section 4 you said that, unlike Tabacof and Costabello, you use all available negatives. This way, you would avoid the sampling bias. Did you measure the effect of such bias compared with your solution? Is the elimination of this bias the only reason to avoid such a sampling?

2. In Section 6.1 you said that the size of the validation split and the variety of the triples contained in it were detrimental to learn calibration functions. I am not sure what do you mean by variety of the triples. Regarding the size, you decided to use validation and test splits to contain 1.6% of the triples each. Does this mean that this method protocol be non-reliable for larger sizes? Does in mean that for this method could not be used in production for datasets with many missing triples?

**Reviewer Confidence:**

1: The reviewer's evaluation is an educated guess

**Scope:**

4: The work is relevant to the Web and to the track, and is of broad interest to the community

---

### Official Review · Reviewer_UnCj · 2023-12-01

**Novelty:** 5
**Technical Quality:** 4

**Review:**

## Summary

This work tackles the problem of evaluating link prediction in knowledge graphs for approaches based on knowledge graph embeddings (KGE). Existing evaluation protocols mainly rely on ranking-based metrics (e.g., hits@k, mean rank, or mean reciprocal rank) that require generating a large number of possible candidates for a triple, i.e., generating positives and negatives. These metrics introduce two major drawbacks to the evaluation protocol: (1) they are computationally expensive, and (2) assessing the membership of a triple to the set of positives is not straightforward as a ranking is required. To overcome these shortcomings, this work proposes a novel protocol that alleviates the computation of rankings during evaluation by learning a calibration model over the scoring function of the KGE model. The model is learned using the full set of negatives — generated with different strategies such as Global Naive (GN), Typed Constraint LCWA (TC), and Local Naive (LC)) — and using Platt scaling or Isotonic regression. The paper also introduces the metrics to assess the quality of the calibration model — i.e., the weighted Brier score BS_w and the weighted coefficient of determination R^2_w — and metrics to compare these values with ranks — i.e., the Pearson correlation r_{xy} with adjusted ranks. The experiments on eight datasets and nine KGE models show that the proposed protocol produces results that slightly correlates with ranking-based approaches and the evaluation time can be reduced up to 98.9%.

## Strong Points

S1. This work tackles a relevant, timely problem for knowledge graphs and link prediction approaches based on **machine learning**.

S2. The proposed evaluation protocol is well-motivated. The authors clearly state the limitations of the state of the art (i.e., rank-based metrics), and explain how the presented solution overcomes the current limitations.

S3. The experimental evaluation includes several knowledge graph embeddings to show the behavior of the new protocol over different models.

## Weak Points

W1. The presentation of the paper requires major improvement. Unfortunately, the structure of the paper and presentation of the experimental results make it difficult to assess the soundness of the overall contribution.

W2. The robustness of the evaluation protocol is sensitive to the learning of the calibration model, whose quality can be impacted by factors (e.g., the dataset, the strategy for generating negatives, the calibration function, etc.) independent of the link prediction model. This could lead to misleading conclusions about the quality of the link prediction approaches.

W3. The interpretability of the new metrics is not sufficiently analysed.

W4. The reproducibility of this work is unknown.

## Detailed Comments

- **Presentation of the paper:** Each section is well-written; however, the overall paper structure is hard to follow. Below are just a few examples of how the different pieces of the work are scattered in different sections:
    - The concept of model calibration is introduced in Section 2.2. as part of the background. Here, only two calibration functions for model calibration, i.e., Platt scaling and isotonic regression, are defined, which are the ones investigated in the paper. Note that there might be other techniques (e.g., Beta calibration) that are not discussed in the paper. So, this section is a mixture of background and proposed solutions simultaneously.
    - Section 3 presents a discussion that refers to Equation 2, but this is coming too late as Section 2 finishes with Equation 9 with the definitions of TP, FP, TN, and FN. The flow of the paper is broken here.
    - Section 5 explains the metrics used to assess the calibration model. Yet, two metrics have already been introduced in the Background (concretely, Section 2.2), and now a new metric, BA, is introduced here. In addition, Section 5 includes another metric to compare the calibration model to rank-based metrics. Is this truly part of the proposed approach? Or is this a metric specific to the experimental study of this paper, which should instead be defined in Section 6?

- **Soundness of the proposed solution:** As mentioned before, it is difficult to assess the soundness of the proposed evaluation protocol, as the pieces are not presented coherently. Unfortunately, the presentation of the experimental results is also hard to follow. Please see the detailed comments below:
    - The paper does not sufficiently discuss the limitations of the proposed protocol, for example, that it is sensitive to the learning of the calibration model. Furthermore, there are no guidelines on which calibration functions, negative generation techniques, etc., to apply in the future to obtain meaningful results for an evaluation.
    - Equation 17: The paper does not explain the choice of the arithmetic mean over other means, for example, the harmonic mean.
    - Figures 1 and 2 show the results for the best calibration function, but it would be interesting to report the results for each calibration function investigated. Without these levels of detail, it is impossible to understand the impact of the different “components” of the new protocol on the observed results.
    - Table 1 shows the time difference (in %) between the proposed and existing solutions. However, the time difference is defined as the sum of the times of the approaches, normalized by the time of the link prediction approach. Why the sum and not the difference? This might lead to values higher than 100%. Table 1 could be simplified by presenting the compared approaches' raw times.
    - Table 2 is also hard to follow.

- **Interpretability of the metrics:**
    - One important aspect when developing a novel benchmarking protocol is to demonstrate that the results obtained with the new techniques can be easily interpreted, i.e., to confirm that the new protocol is behaving as expected. The paper includes several passages about this matter, but this is not demonstrated with the experimental results.
    - The paper does not show how the proposed protocol scores individual triples using BA, BS_w, or R^2_w. This was one of the limitations of the state of the art discussed in the introduction. It would be great if the paper provides examples of how this is achieved with the new protocol.

- **Reproducibility:**
    - Footnote 4 indicates that the required sources to reproduce the results will be available after the review. But at the time of review, it is not possible to assess how easy the reproducibility of this work is: does the repository include all the necessary files? does the repository include a well-documented README with instructions to repeat the experiments?
    - The authors may consider using services like Anonymous Github (https://anonymous.4open.science) for future submissions. This allows reviewers to assess the reproducibility of the work without compromising the double anonymization.

- **Relevance to the Web:**
    - This work perfectly fits the topic of “knowledge graphs”. Yet, the connection of this paper to the overarching scheme of the conference, i.e., The Web, is not straightforward.
    - This type of work is more suitable for a machine learning or representation learning conference.
    - This remark does not (negatively) influence the overall rating of this paper, but it is more of a suggestion for fitting venues for this work.

**Questions:**

Q1. About the sensitivity of the proposed evaluation protocol, how do the different factors (dataset, generation of negatives, calibration function, etc.) affect the robustness of the results obtained with the new protocol?

Q2. In Equation (17), why not use the harmonic mean between TPR and TNR (i.e., which resembles the F-measure between precision and recall) instead of the arithmetic mean?

Q3.  Can you provide concrete examples of how triples are scored with the new protocol and compare them to the ranking obtained with a rank-based metric?

Q4. Do you have any concrete guidelines on configuring the proposed protocol to ensure high robustness and interpretability of the results?

**Reviewer Confidence:**

4: The reviewer is certain that the evaluation is correct and very familiar with the relevant literature

**Scope:**

3: The work is somewhat relevant to the Web and to the track, and is of narrow interest to a sub-community

---

### Decision · Program_Chairs · 2024-01-22

**Decision:**

Accept

**Comment:**

The paper is sound in methods and experiments, as acknowledged by the reviewers. The issues regarding methodology and fitness to the conference have been addressed, hence the paper evaluation has also increased toward weak acceptance.